# Time- and polarity-dependent proteomic changes associated with homeostatic scaling at central synapses

Christoph T Schanzenbächer[1,2], Julian D Langer[1,2]*, Erin M Schuman[1]*

[1]Max Planck Institute for Brain Research, Frankfurt am Main, Germany; [2]Max Planck Institute of Biophysics, Frankfurt am Main, Germany

**Abstract** In homeostatic scaling at central synapses, the depth and breadth of cellular mechanisms that detect the offset from the set-point, detect the duration of the offset and implement a cellular response are not well understood. To understand the time-dependent scaling dynamics we treated cultured rat hippocampal cells with either TTX or bicucculline for 2 hr to induce the process of up- or down-scaling, respectively. During the activity manipulation we metabolically labeled newly synthesized proteins using BONCAT. We identified 168 newly synthesized proteins that exhibited significant changes in expression. To obtain a temporal trajectory of the response, we compared the proteins synthesized within 2 hr or 24 hr of the activity manipulation. Surprisingly, there was little overlap in the significantly regulated newly synthesized proteins identified in the early- and integrated late response datasets. There was, however, overlap in the functional categories that are modulated early and late. These data indicate that within protein function groups, different proteomic choices can be made to effect early and late homeostatic responses that detect the duration and polarity of the activity manipulation.

DOI: https://doi.org/10.7554/eLife.33322.001

**\*For correspondence:**
julian.langer@biophys.mpg.de
(JDL);
erin.schuman@brain.mpg.de
(EMS)

**Competing interests:** The authors declare that no competing interests exist.

## Introduction

In biological systems, many variables like pH, ion concentration and temperature are under homeostatic control. In the brain, changes in synaptic efficacy underlie at least some forms of information storage, potentially leading to runaway dynamics of synaptic weights and activity. Brain synapses also exhibit homeostasis. Synaptic scaling, a form of homeostatic plasticity, allows neurons to adjust globally the strengths of their synapses up or down to stabilize specific neuronal functions in response to an offset from a setpoint (*Turrigiano et al., 1998*; *Davis and Goodman, 1998*; *Burrone et al., 2002*; *Turrigiano, 2012*). Synaptic scaling has been observed and studied at central nervous system synapses in the hippocampus and cortex as well as the neuromuscular junction (see *Davis, 2013* and *Turrigiano, 2008* for review).

The cascade of mechanisms that underlie the homeostatic response are not well understood. Mechanistically speaking, there must exist molecules and signaling events that detect the offset from the activity set-point and implement cellular responses to maintain homeostasis. Several studies have suggested that activity-dependent changes in intracellular $Ca2+$ may serve as an early sensor for the offset from an activity threshold (*Ibata et al., 2008*; *Thiagarajan et al., 2002*, *2005*). In homeostatic scaling at central mammalian synapses, there is emerging consensus that one end-point effector mediating both the enhanced and reduced response following a prolonged activity manipulation is the AMPA-type glutamate receptor (*Cingolani et al., 2008*; *O'Brien et al., 1998*; *Wierenga et al., 2005*; *Thiagarajan et al., 2005*; *Sutton et al., 2006*; *Gainey et al., 2009*, *2015*).

**eLife digest** The brain can store information by changing the strength of connections between neurons, also known as synapses. When two neurons at a synapse are active at the same time, the synapse becomes stronger. This enables the first neuron to activate the second more easily. But it also means that the two neurons will now be active at the same time more often, which will tend to make the synapse even stronger. If this process continues unchecked, the synapse will keep getting stronger until no further changes in strength are possible. This will make it harder for the brain to form new memories.

To prevent this from happening, the brain responds to prolonged changes in the activity of neurons by adjusting the strength of synapses in the opposite direction. If neurons are too active for an extended period of time, the brain reduces the strength of synapses. If neurons show too little activity, the brain increases the strength of synapses. This process is known as homeostatic scaling, and the brain achieves it by adjusting the number and/or type of proteins present at synapses.

Schanzenbächer et al. now reveal the changes in synaptic proteins that occur in response to a two-hour increase or decrease in neuronal activity. These changes can be tracked in the laboratory by growing cells in a petri dish in the presence of modified amino acids, the building blocks of proteins. Any new proteins the cells produce will contain the modified amino acids, making them easy to spot. Schanzenbächer et al. applied this technique to neurons obtained from the rat hippocampus, a region of the brain involved in learning and memory. Bathing the neurons for two hours in chemicals that either enhanced or reduced their activity, triggered changes in more than 150 proteins.

Schanzenbächer et al. compared these results to those of a previous experiment in which neuronal activity had been manipulated for 24 hours. Each set of conditions produced a characteristic profile of protein activity. The profiles indicated whether the activity in neurons had increased or decreased, and whether the changes had lasted for two hours or 24 hours. These findings may provide insights into disease states in which there is too much or too little brain activity.

DOI: https://doi.org/10.7554/eLife.33322.002

At the systems level, homeostatic scaling requires both transcription (*Ibata et al., 2008*) and translation (*Schanzenbächer et al., 2016*). Global activity manipulations bring about both transcriptional (*Meadows et al., 2015*; *Steinmetz et al., 2016*; *Schaukowitch et al., 2017*) and translational regulation (*Schanzenbächer et al., 2016*). For example, chronic activity suppression activated a calcium-dependent transcription program of 73 genes that includes AMPA receptors and transcription factors (SRF, ELK) and led to synaptic upscaling (*Schaukowitch et al., 2017*). After 24 hr of either enhanced or reduced activity, *Schanzenbächer et al., 2016* detected 307 differentially expressed proteins involved in neuronal functions such as calcium transport, synaptic vesicle trafficking and neurotransmitter release.

The earliest reports of homeostatic scaling at mammalian central synapses suggested that relatively long duration activity manipulations were required to elicit the homeostatic response (*O'Brien et al., 1998*; *Turrigiano et al., 1998*). However, local manipulations of activity can bring about more rapid scaling (*Sutton et al., 2004*, *2006*) and recent studies have also suggested that global scaling can occur within a few hours of an activity manipulation (*Ibata et al., 2008*). Under physiological conditions that elicit synaptic scaling, it is likely that different durations of activity manipulation give rise to different responses that may differ both in the magnitude and nature of regulated proteins. However, the mechanisms that enable neurons to sense the duration and polarity of a stimulus remain poorly understood to date.

In order to understand the time-dependent dynamics of the response to an activity manipulation we treated cultured hippocampal cells with either TTX or bicuculline for 2 hr to induce the process of up- or down-scaling (*Turrigiano et al., 1998*), respectively. During the 2 hr activity manipulation we metabolically labeled newly synthesized proteins using BONCAT (*Dieterich et al., 2006*; *Schanzenbächer et al., 2016*). We analyzed the newly synthesized proteins and then to obtain a temporal trajectory of the cellular response, we compared the proteins synthesized within 2 hr of

activity manipulation to those synthesized during a 24 hr manipulation (*Schanzenbächer et al., 2016*). Surprisingly, there was little overlap in the significantly regulated newly synthesized proteins identified in the early- and integrated late response datasets. Many significantly regulated at 24 hr, however, exhibited trends for intermediate levels of regulation at 2 hr. There was however, overlap in the functional categories that are modulated at the two time points, suggesting that within protein function groups, different choices are made to effect early and late homeostatic responses.

## Results

In order to determine the early cellular response to neural activity manipulations, we treated cultured hippocampal cells with either TTX or bicuculline for 2 hr to induce the process of up- or down-scaling (*Turrigiano et al., 1998*) respectively, and during this time metabolically labeled newly synthesized proteins using azidohomoalanine (AHA) and BONCAT (*Dieterich et al., 2006*; *Schanzenbächer et al., 2016*). We compared the newly synthesized proteome in the TTX- and biculline-treated samples to each other and to an untreated control group, analyzing five biological replicates. Proteins were identified and quantified using MaxQuant (*Cox and Mann, 2008*), with the following criteria: (i) at least one peptide per protein identified (ii) protein detected in at least 1/5

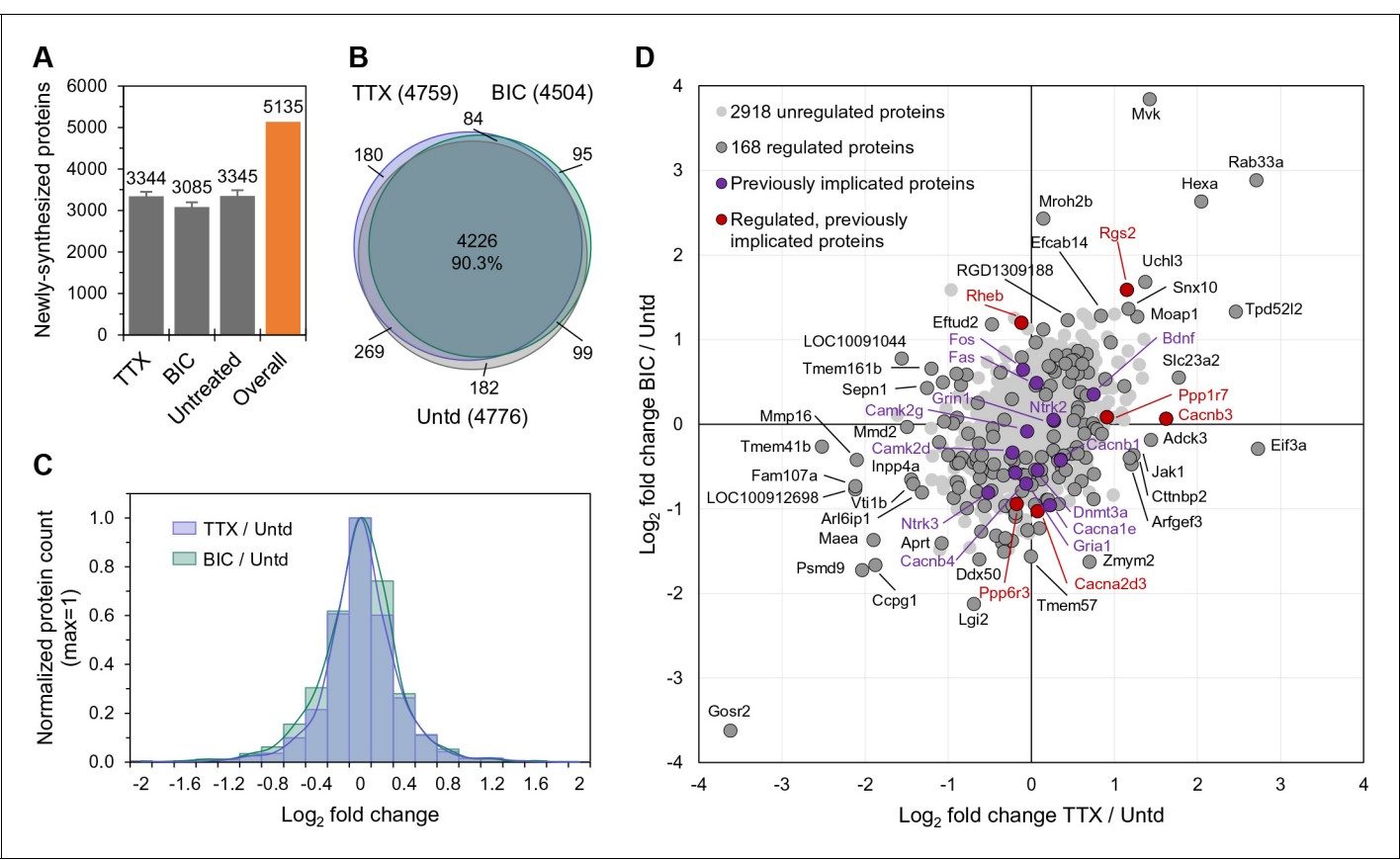

**Figure 1.** The newly synthesized proteome after 2 hr of enhanced or reduced activity. (**A**) Bar chart showing the number of newly synthesized proteins identified on average, in each group as indicated. Error bars represent ±SEM of 5 biological replicates. (**B**) Venn diagram indicating the number (in parentheses) and overlap of proteins expressed in each treatment group (TTX or bicuculline) and untreated control. (**C**) Protein fold changes ($Log_2$) showing the overall proteome regulation each treatment group compared to untreated samples. (**D**) Scatter plot showing all significantly regulated proteins (dark grey; n = 168; ANOVA, FDR = 0.05) as well as the regulation of proteins previously implicated in homeostatic scaling (red). Proteins shown in light grey or purple are not significantly regulated.

DOI: https://doi.org/10.7554/eLife.33322.003

The following figure supplement is available for figure 1:

**Figure supplement 1.** Data reproducibility and overlap of the newly synthesized proteome (2 hr).

DOI: https://doi.org/10.7554/eLife.33322.004

(*Figure 1A–B*) or 2/5 (all subsequent Figures) experiments and (iii) protein not detected in the Met control group (see Materials and methods). The methionine content of our AHA-labeled proteins was not different compared to a global proteome (*Rattus norvegicus*, UniProtKB database) as, on average, 2.42 and 2.40 Met residues are present per 100 amino acids, respectively (*Figure 1—figure supplement 1I*). As shown in *Figure 1A*, we identified over 3000 proteins in each treatment group and over 5000 unique proteins altogether (summing proteins across groups, see *Supplementary file 1*). The majority of proteins were identified in at least 3/5 replicates and the relative representation of proteins in different functional groups as well as the abundance of proteins was not different between conditions (*Figure 1—figure supplement 1A–C*). The protein intensities associated with replicates within each treatment group were well-correlated with one another as were the protein intensities for each 2 hr treatment group (*Figure 1—figure supplement 1D–G*). As observed previously (*Schanzenbächer et al., 2016*) the activity manipulations did not result in a significant change in the overall proteome size (*Figure 1A*). Indeed, the overlap in the proteins identified in the three groups was greater than 90% (*Figure 1B*). The newly synthesized proteins uniquely identified in each group were 180, 95 and 182 for TTX-, bicuculline- and the untreated groups, respectively, representing about 3–6% of each total nascent proteome. We analyzed these unique proteins and found that the majority of them were detected in a single replicate (*Figure 1—figure supplement 1H*). A small number of proteins (4, 2, and 3 for TTX, bicuc, and untreated, respectively) were detected in 3/5 replicates and represented cytoskeletal-associated proteins and signaling molecules (*Figure 1—figure supplement 1H*).

Neither the stimulation (bicuc) nor inhibition (TTX) of activity induced a systematic shift (e.g. global up- or down-regulation) of proteins (*Figure 1C*), but rather induced either up- or down-regulation of different sets of proteins. We focused our analysis on these changes in the level of protein expression, using label-free quantification (LFQ) (*Cox et al., 2014*) to analyze the intensities of identified peptides in different groups (*Figure 1—figure supplement 1C*). Altogether, we detected 168 proteins whose expression level was significantly regulated by either enhanced or reduced activity or by both manipulations (*Supplementary file 2*). In *Figure 1D*, proteins significant up- or down-regulated by both enhanced and reduced activity are shown in the upper-right and lower-left quadrant, respectively. Note that the regulation of proteins in these two quadrants does not give information about the direction of the activity manipulation (e.g. up or down, the 'polarity') but rather could signal the absolute value of the deviation from an activity set-point. Other proteins exhibited significant increases in expression levels that were specific to either enhanced (upper-left quadrant) or reduced (lower-right quadrant) activity (*Figure 1D*). In principle, these classes of regulated proteins could detect or represent both the offset from the setpoint and the sign of the offset. We compared the regulated proteins to a curated list of 35 proteins previously implicated in homeostatic scaling by other groups (*Supplementary file 2*; *Schanzenbächer et al., 2016*). We found six proteins regulated in our dataset that have been previously implicated in scaling, including two Ca2+ channel subunits (Cacnb3 and Cacna2d3), two protein phosphatases (Ppp1r7 and Ppp6r3), and two G-protein family members (Rheb and Rgs2).

Which functional protein classes exhibit significant differential regulation following 2 hr of enhanced or reduced activity? We conducted a gene annotation (GO) analysis of the 168 significantly regulated proteins and found that most (152/168) regulated proteins had a neuronal function and many were associated with synaptic functions (*Figure 2A*, *Supplementary file 3*), indicating that the majority of the significantly regulated proteins are not simply associated common cellular or metabolic functions. Using the newly synthesized proteins in untreated samples as a background, we examined which particular functional groups were regulated by the activity manipulations (*Figure 2B*, *Supplementary file 3*). Amongst the top 20 terms that showed significant enrichment were many terms associated with the secretory pathway, including Golgi apparatus, ER to Golgi transport vesicle, and Golgi membrane. Within the synapse, we noted the regulation of many proteins including Neuroligin 2 (Nlgn2), Neurexin 2 (Nrxn2) Agrin (Agrn) and Slc32a1 (*Figure 2C*).

Does the up- or downward manipulation of activity result in protein groups which are differentially regulated, reflecting the 'sign' of the manipulation? We found many protein groups that exhibit significant differential regulation between the activity manipulated groups (*Figure 3*, *Supplementary file 4*). For example, the protein functional groups significantly upregulated following TTX-treatment (and down-regulated following bicuculline-treatment) included the following categories: voltage-gated Ca2+ channels, synaptosome, dendritic spine as well as cell adhesion, actin

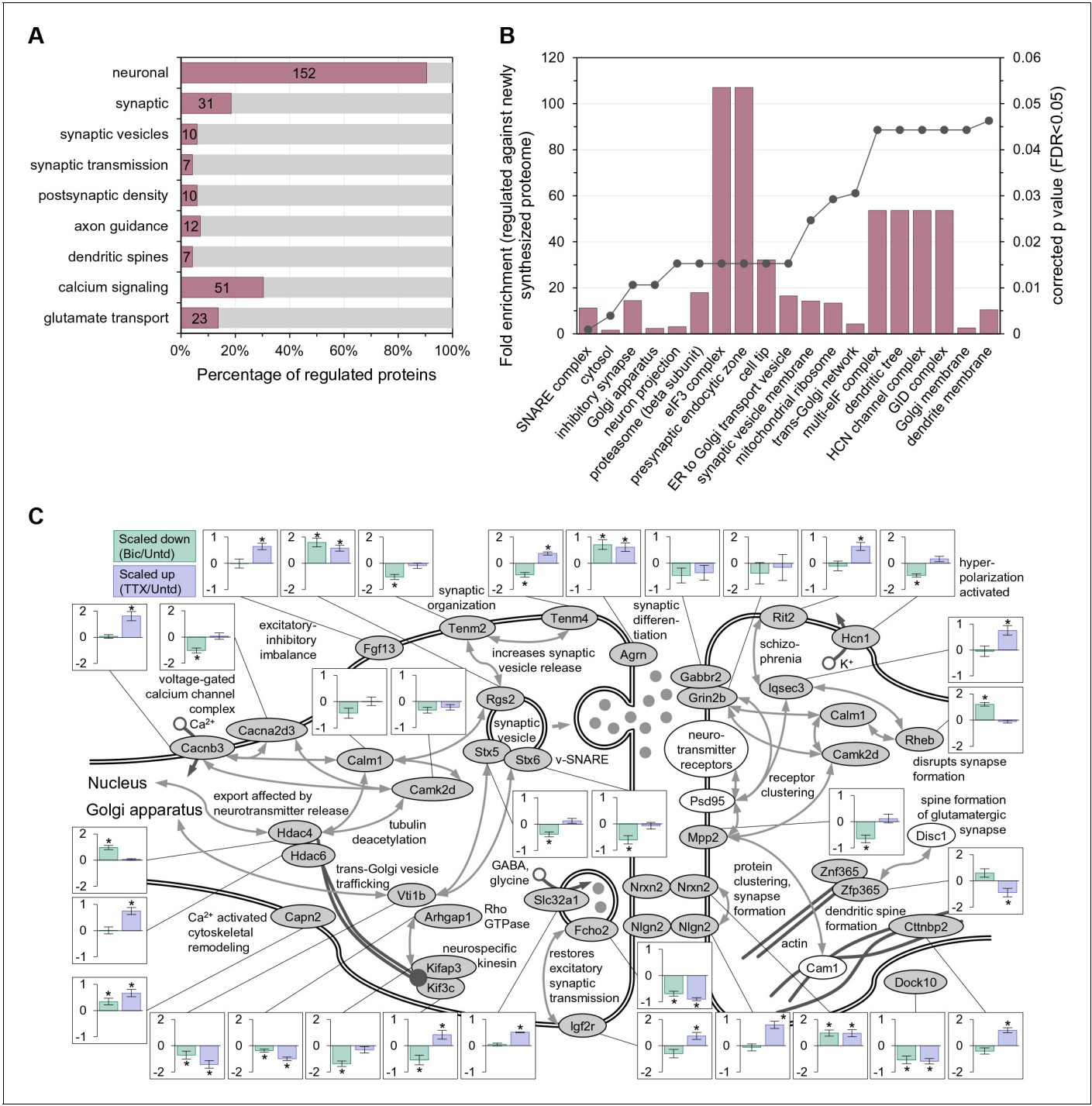

**Figure 2.** Enriched function groups for proteins that exhibited significant regulation after 2 hr of enhanced or reduced activity. (**A**) Gene Ontology enrichment categories for all 168 regulated proteins. A majority of the proteins were associated with the neuronal function group. Note that individual regulated proteins can belong to more than one group. (**B**) Cellular function enrichment analysis showing the fold-enrichment for the indicated groups (left y-axis, bars) as well as the corrected p value (right y-axis, line). (**C**) Significantly regulated proteins associated with the synapse and nuclear function. Bar graph inserts show the regulation for scaling-down (bicuculline treatment; pale green bars) or scaling-up (TTX treatment; lavender bars).
DOI: https://doi.org/10.7554/eLife.33322.005

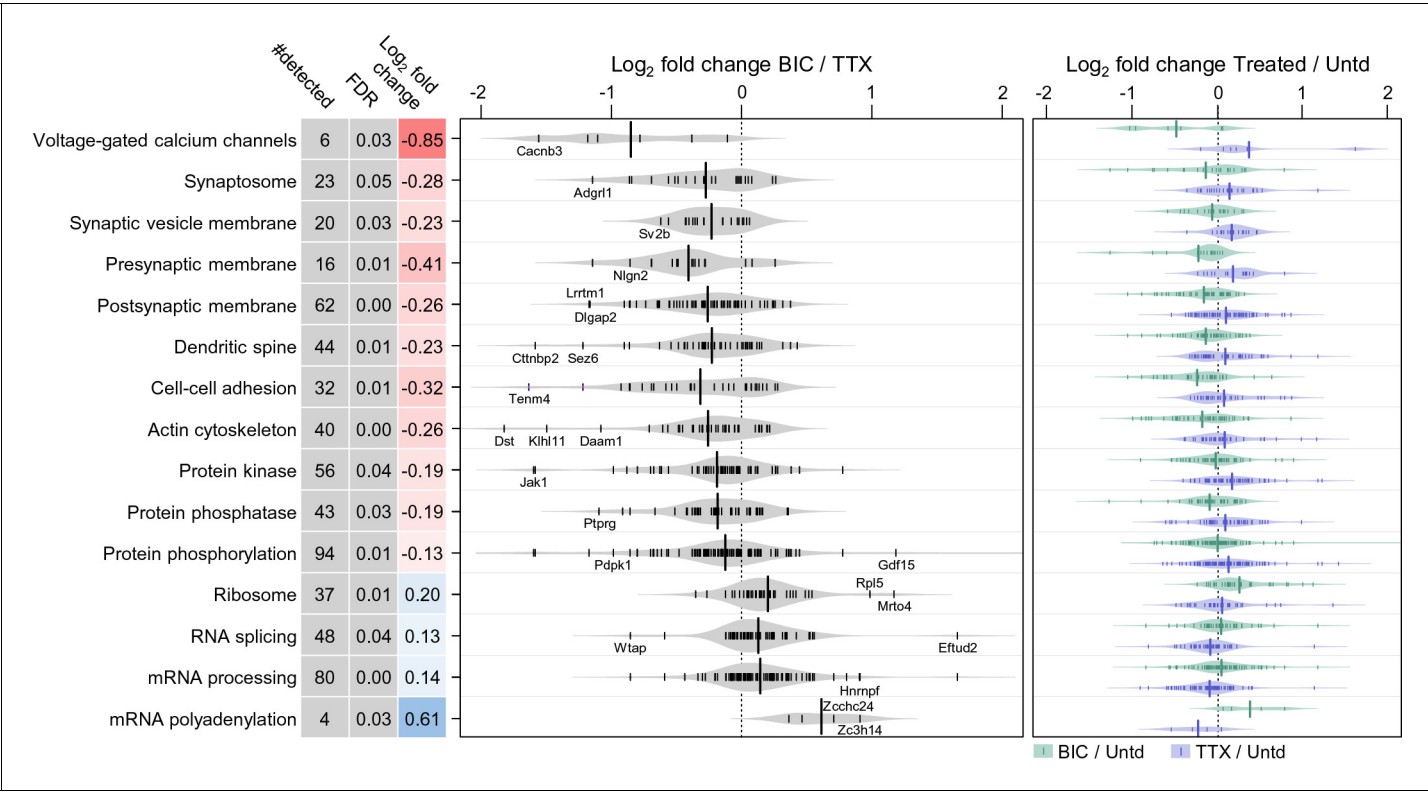

**Figure 3.** Polarized differential regulation of some protein functional classes after 2 hr of enhanced or reduced activity. Bean plot of selected significantly regulated protein classes and pathways that exhibited polarized regulation after 2 hr treatment with either TTX or bicuculline. #detected: number of proteins assigned to specified group, FDR: false discovery rate, $Log_2$ fold change: intensity ratio of bicuculline vs TTX treated neurons. Proteins (black markers) were grouped into functional classes and pathways as described in Materials and methods. The bean plots in grey show the comparison between bicuculline and TTX, the green and blue plots show the comparison of each treatment group to a control.
DOI: https://doi.org/10.7554/eLife.33322.006

cytoskeleton and proteins associated with phosphorylation and de-phosphorylation. At the other end of the spectrum were several protein groups associated with protein translation, that were negatively regulated by TTX treatment and positively regulated by bicuculline (*Figure 3*). Some examples of polarity-dependent regulation of proteins are also shown in *Figure 2C*, for example, the kinesin Kif3c, the vesicle-associated protein syntaxin 5 (Stx5), the hyperpolarization-activated channel Hcn1, and the disease-related protein Teneurin transmembrane protein 4 (Tenm4).

How does the proteomic response to an activity deviation evolve over time? To address this, we compared the significantly regulated nascent proteins following 2 hr of activity manipulation (either bicuculline or TTX) to those regulated during a 24 hr manipulation (initially described in *Schanzenbächer et al., 2016*). In the absence of activity manipulation, the newly synthesized proteomes identified at these two time points were largely (~94%) overlapping (*Figure 4A* and *Figure 4—figure supplement 1A*). There were 292 (2 hrs) and 1287 (24 hrs) proteins that were uniquely identified at one time point, but not the other. Using peptide intensities, we analyzed the relative abundance of these unique proteins compared to the total pool and found that they were less abundant (*Figure 4—figure supplement 1A*), suggesting that the inability to detect them in one group might be due to insufficient sensitivity. We next analyzed the significantly regulated proteins at each time point (168 at 2 hr and 307 at 24 hr) and found a very low level of overlap (10 proteins) (*Figure 4B* and *Figure 4—figure supplement 1B*, *Supplementary file 5*), not significantly greater than what would be expected by chance. As also noted in *Figure 1D*, a large fraction of proteins was significantly regulated by both TTX and bicuculline; at 2 hr 52% of the regulated proteins exhibited regulation by both treatments (*Figure 4C*). We noted with interest that the fraction of proteins that were uniquely regulated by either TTX or bicuculline was enhanced (*Figure 4D*), suggesting

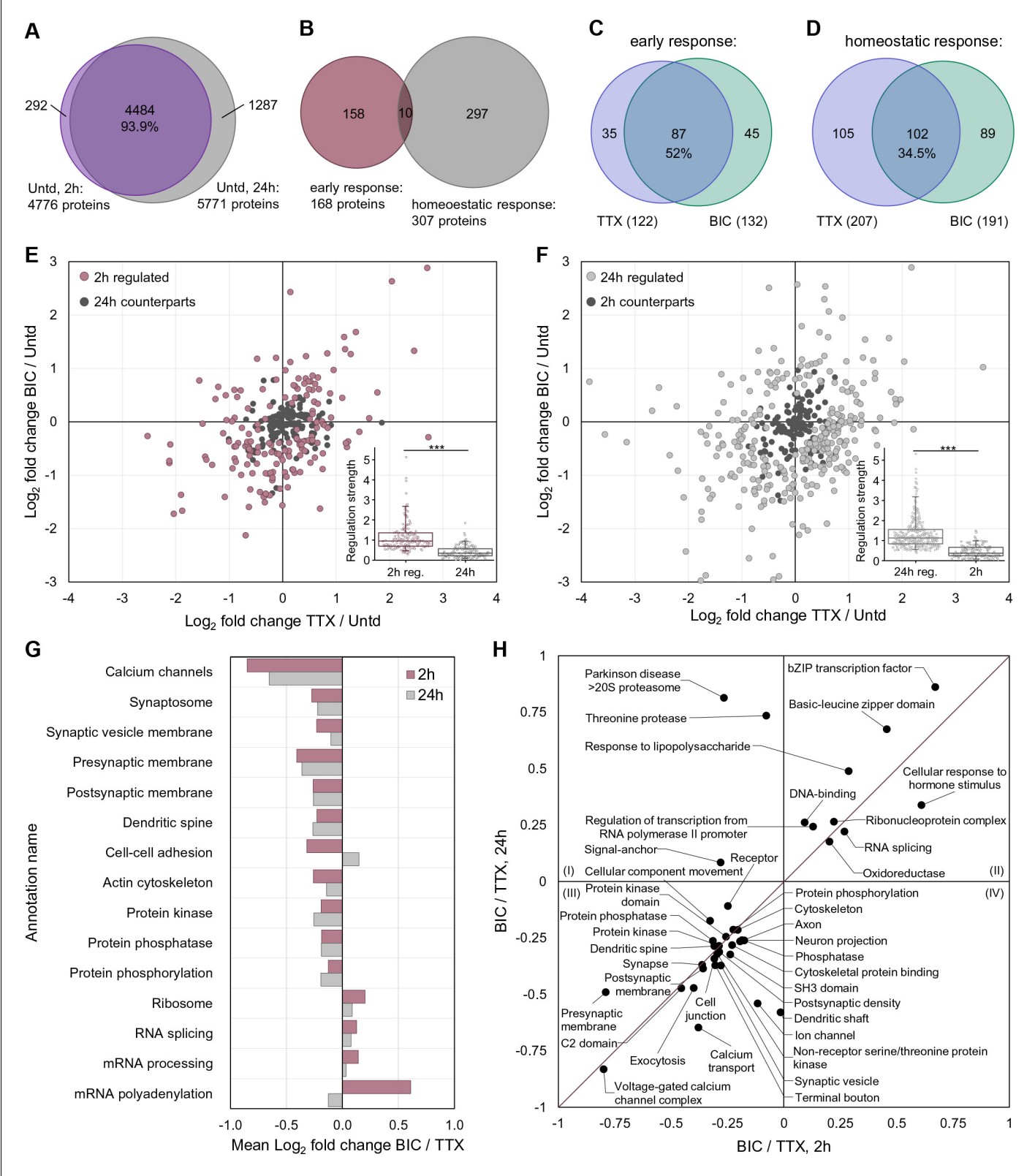

**Figure 4.** A comparison of proteome remodeling in neurons after 2 hr and 24 hr of synaptic scaling. (**A**) Venn diagram showing the overlap of proteins expressed in untreated neurons for 2 hr and 24 hr. (**B**) Venn diagram showing the overlap of the significantly regulated proteins following 2 hr or 24 hr of synaptic scaling. (**C**) Venn diagram showing the overlap of the proteins significantly regulated by TTX and bicucculine treatment after 2 hr. The total number of regulated proteins for each group is in parentheses. (**D**) Venn diagram showing the overlap of the proteins significantly regulated by TTX

*Figure 4 continued on next page*

*Figure 4 continued*

and bicucculine treatment after 24 hr. The total number of regulated proteins for each group is in parentheses. (E) Scatter plots showing the regulated proteins at 2 hr (mauve dots) and their corresponding regulation at 24 hr (black dots). Inset shows the median regulation strength for the same set of proteins. (F) Scatter plots showing the regulated proteins at 24 hr (grey dots) and their corresponding regulation at 2 hr (black dots). Inset shows the median regulation strength for the same set of proteins. (G) Functional annotation of protein groups showing that, although overlapping individual regulated proteins are rare at 2 and 24 hr, there are similar protein groups that exhibit differential regulation between the up- and down-scaling treatment. (H) More detailed functional characterization of the protein groups that are similarly regulated at 2 and 24 hr.

DOI: https://doi.org/10.7554/eLife.33322.007

The following figure supplement is available for figure 4:

**Figure supplement 1.** Further analyses of 'unique' proteins identified in untreated samples at 2 and 24 hr.
DOI: https://doi.org/10.7554/eLife.33322.008

that as the duration of the treatment increases, the distinctiveness of the proteomic response, presumably more focused on the homeostatic effector mechanism, evolves.

We next focused our analyses on the translational dynamics of the proteins which showed significant regulation at either the 2 or 24 hr time point. We examined the significantly regulated proteins for each time point and then examined the same protein at the other (not significantly) regulated time point. In *Figure 4E*, the mauve dots represent the proteins regulated at 2 hr, with proteins up- and down-regulated following both enhanced (bicuculline-treated) or reduced (TTX-treated) activity shown in the upper-right and lower-left quadrants, respectively. The black dots in this plot show the regulation of these same proteins at 24 hr. The clustering of the black dots near the origin indicates that the newly synthesized proteins significantly regulated at 2 hr are largely unregulated at 24 hr. In *Figure 4F*, the grey dots represent the proteins regulated at 24 hr and the black dots represent these same proteins at 2 hr. Again, the cluster of black dots around the origin thus indicates the absence of significant regulation of these proteins at the 2 hr time point. Although there was little evidence that the same individual proteins were regulated at both 2 and 24 hr following either activity manipulation, we considered the possibility that a functional protein group might be regulated at both time points, with the individual regulated proteins within that group being different at 2 vs 24 hr. Indeed, we found several examples where there was significant regulation of a functional protein group at both time points, although the individual proteins within the group were different at each time point (*Figure 4G and H*, *Supplementary file 4*). For example, the group 'voltage-gated calcium channels' was significantly regulated at both 2 and 24 hr, but the individual subunits were different at the time

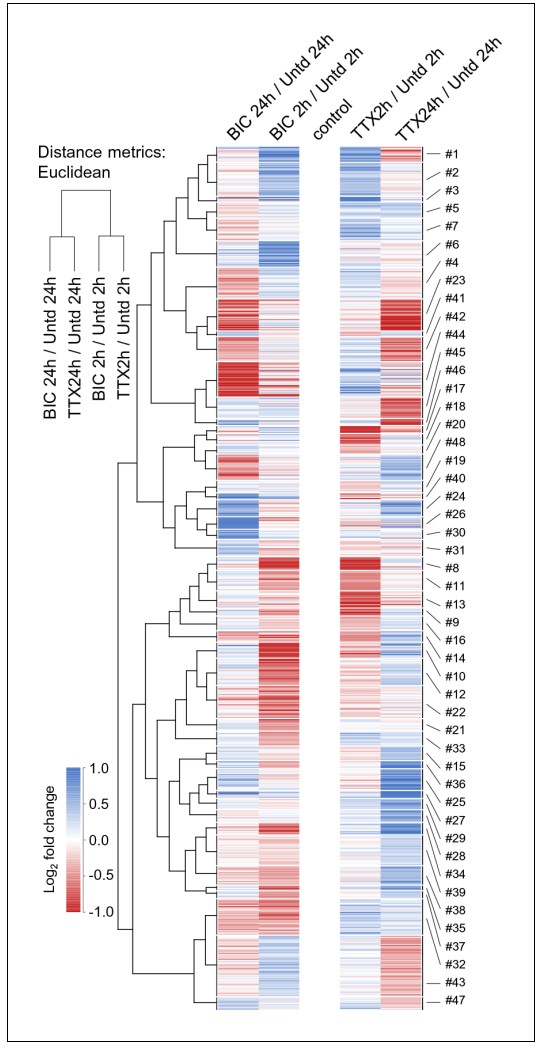

**Figure 5.** Time-dependent protein regulation in synaptic scaling: patterns of time- and polarity-dependent regulation. Heat-map showing the clustered patterns of time- and polarity-dependent regulation, for each type of treatment (TTX or bicucculine) and for both (2 and 24 hr) timepoints. The self-organizing maps corresponding to these clusters are shown in *Figure 6*.
DOI: https://doi.org/10.7554/eLife.33322.009

points (e.g. Cacnb3 and Cacna2d3 at 2 hr and Cacnb4 at 24 hr). In *Figure 4H*, the protein groups that fall on the diagonal line are regulated in the same direction at both the 2 and 24 hr time points, including again, synaptic transmission, dendritic spine, synapse, postsynaptic density, axon and others. Only a few groups, including threonine kinase and Parkinson's disease 20S proteasome were exclusively regulated at 24 hr (*Figure 4H*). These data indicate that most of protein groups are hot-spots for regulation at both time points, even if the individual regulated proteins within the groups are different.

We next conducted a meta-analysis of all significantly regulated proteins considering, in addition to newly synthesized proteins that are different from untreated samples, proteins that are significantly different between the two time-points and/or the up- and down-scaling conditions. This master set of comparisons resulted in 711 significantly regulated proteins, which were significantly enriched for neuronal, rather than glial markers (*Supplementary file 6*). We examined the regulation of each protein at 2 hr and 24 hr for both bicuculline- and TTX-treated samples to discover patterns of co-regulation (*Figures 5* and *6*). We used self-organizing maps (SOM) to sort and organize the various patterns of regulation of individual proteins into similar groups, calculated the average fold-change for each cluster, and then used hierarchical clustering to examine the relationships between the groups (*Figure 5*). In order to extract repeated patterns of regulation, we plotted the individual regulated proteins as a deviation from untreated, including the response at both time points and in both conditions on a single axis (*Figure 6*, *Supplementary file 7*). We first evaluated the optimal number of clusters to use, optimizing a minimization of within-cluster variance and a maximization of between-cluster variance (*Figure 6—figure supplement 1A*). Based on these considerations we chose a cluster size of 48 with each cluster containing between 4–53 proteins, (*Figure 6*, *Supplementary file 7*). The SOMs revealed several distinct patterns of regulation that, in principle, can serve as indicators of either the duration (early-2hrs or late-24hrs) or the polarity (enhanced or reduced) of the activity manipulation. We also analyzed whether particular regulatory patterns were enriched for particular functional classes of proteins (*Figure 6—figure supplement 1B*, *Supplementary file 7*). We found that most (11/12) of the regulatory patterns identified by the SOM algorithm were indicators of the time/duration of the manipulation. For example, proteins that were significantly regulated at 2 hr, but not at 24 hr for both up- and down-scaling fall into this category and their regulation pattern is represented by the 'M' or the 'W' profiles in *Figure 6A–D*. There were also many examples of 'sine wave' shape regulated proteins in which the early (2 hr) and the late (24 hr) time point were regulated in an opposite manner (*Figure 6H*). We also observed profiles in which there was regulation of proteins exclusively at the 24 hr time point in one or both conditions ('upward or downward trapezoid shapes') (*Figure 6E,F,K*). The trapezoid patterns were significantly enriched for a variety of channels and transporters (*Figure 6—figure supplement 1B*). In addition, some patterns of regulation that occurred indicate the polarity (sign) of the activity manipulation. Polarity indicators are clusters where regulation of proteins occurs only in response to one (TTX *or* bicuc) of the activity manipulations or where regulation occurs to both activity manipulations but is of an opposite sign (e.g. protein up regulated following TTX but down-regulated following bicuculline and vice versa). We found that many (7/12) of the regulatory patterns identified by the self-organizing map were indicators of the polarity of the manipulation. Examples of polarity-sensitive patterns include the 'sun-seeking worm' (*Figure 6G*; enriched for kinases, *Figure 6—figure supplement 1B*), the 'sine wave' (*Figure 6H*) or the 'skewed W' (*Figure 6I*), 'skewed M' (*Figure 6J*) or 'flattened trapezoid' (*Figure 6K*) profiles. We noted with interest that some simple protein regulation patterns were also apparently absent from our data (*Figure 6L,M*). For example, the sign and magnitude of a single protein's regulation could convey information about both the time and the direction of the activity manipulation, as shown in the 'diagonal line' pattern in *Figure 6L*. This pattern was not, however, observed, although some sub-features of this pattern can be found in some of the other clusters. Alternatively, time- information, but not polarity information, could be conveyed by the 'V' or 'inverted V' (*Figure 6M*); this pattern was also not observed in our data. Taken together, these data indicate that there are patterns of individual protein regulation that can convey information about either the duration of the activity offset (time) or the polarity (up-scaling or down-scaling) of the offset.

Lastly, to examine whether there is a functional relationship between the large set of regulated proteins we conducted a network analysis (see Materials and methods), including any protein significantly regulated at either 2 or 24 hr following either bicuculline or TTX treatment (*Figure 7*), yielding

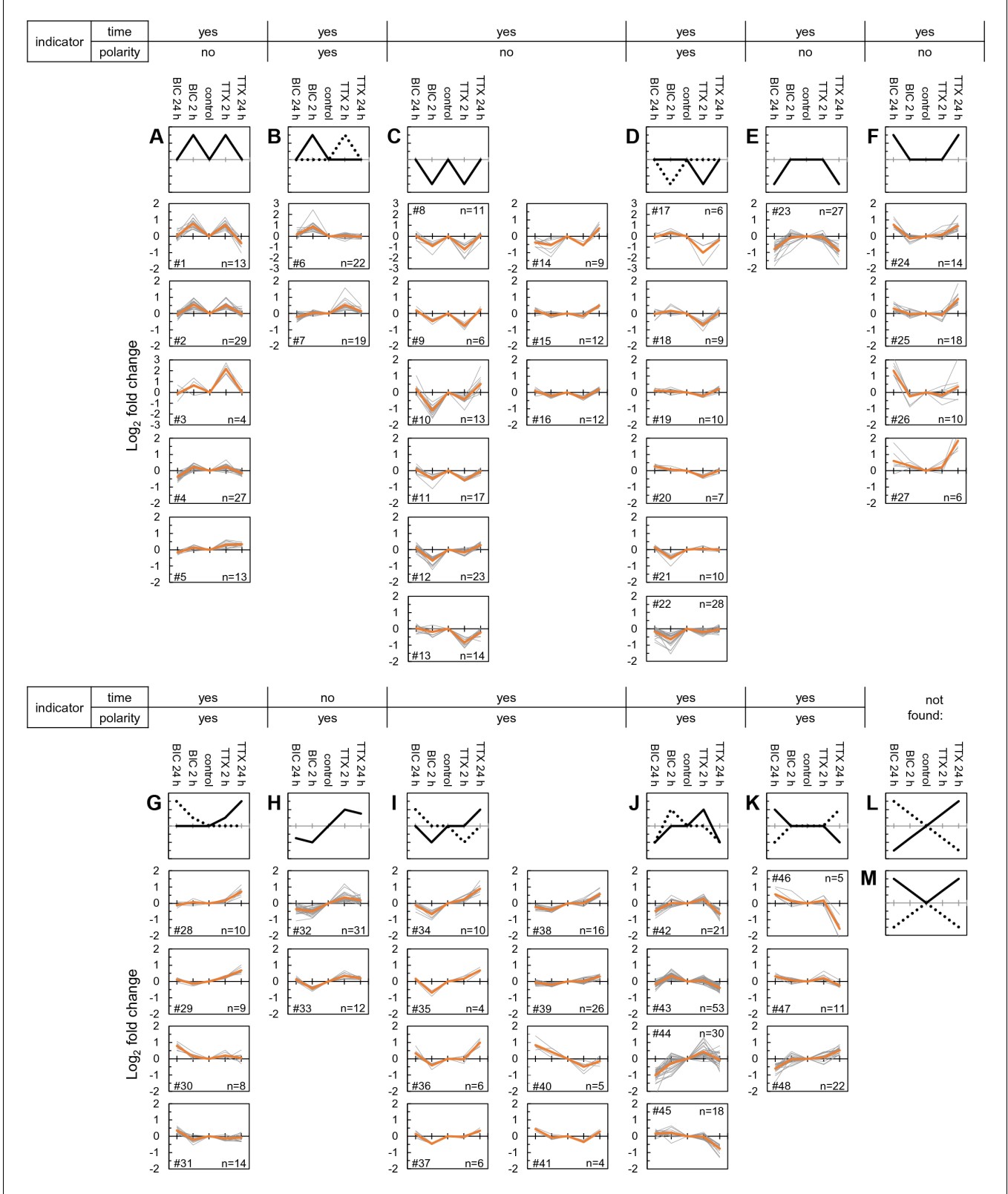

**Figure 6.** Self-organizing maps define clusters of time- and polarity-dependent proteome regulation. (A–M) Shown are clusters of regulated protein expression profiles obtained using self-organizing maps. Along the top of the clusters, whether or not the expression profile indicates the time the system has been manipulated (e.g. 2 or 24 hr) or the type of manipulation (up- or down-scaling). (A) The 'M' profile, represented by five clusters and a total of 86 different proteins. (B) The modified 'M' profile, represented by 2 clusters and 41 different proteins. (C) The 'W' profile, represented by nine

*Figure 6 continued on next page*

*Figure 6 continued*

clusters and a total of 117 different proteins. (D) The modified 'W' profile, represented by 6 clusters and 70 different proteins. (E) The 'inverted trapezoid' profile represented by 1 cluster of 27 proteins. (F) The 'trapezoid' profile represented by 4 clusters and 48 proteins. (G) The 'sun-seeking worm' profile represented by 4 clusters and 41 proteins. (H) The 'sine wave' profile represented by 2 clusters and 43 proteins. (I) The 'skewed W' profile represented by 8 clusters and 77 proteins. (J) The 'skewed M' profile represented by 4 clusters and 122 proteins. (K) The 'flattened trapezoid' profile represented by 3 clusters and 38 proteins. (L and M) Two profiles that we did not identify in our population of newly synthesized proteins include the 'diagonal lines' or the 'regular or inverted V'.

DOI: https://doi.org/10.7554/eLife.33322.010

The following figure supplement is available for figure 6:

**Figure supplement 1.** Optimization of cluster size for self-organizing maps and gene ontology analysis of cluster patterns.

DOI: https://doi.org/10.7554/eLife.33322.011

a regulated protein-protein interaction map. This analysis revealed several protein networks, again comprising functions associated with the cytoskeleton and vesicle-mediated transport (*Figure 7A,B*), neuronal systems (*Figure 7A,C*), cell-cell communication (*Figure 7A,D*), the processing and transport of mRNA (*Figure 7A,E*) and mitochondrial translation (*Figure 7F*). Within these networks signaling molecules, predominantly protein kinases and phosphatases emerged as hubs. Also, two catalytic subunits of the cAMP-dependent Protein Kinase A (Prkaca and Prkacb) display highly correlated changes following both Bic and TTX stimulation with a pronounced abundance increase after 24 hr of TTX and decrease after 2 hr of Bic. Prkaca and Prkacb interact with multiple regulated proteins that often exhibit similar changes in abundance and are known to play essential roles in synaptic transmission and plasticity: The voltage-gated calcium channel auxiliary beta subunits 1, 3, and 4 (Cacnb1, 3, 4) and alpha subunit 2 delta 3 (Cacna2d3), Calmodulin (Calm1) and the calcium-dependent protein kinase 2 subunit delta (Camk2d). Similarly, Synaptojanin-1, a protein that affects synaptic transmission and membrane trafficking by regulating membrane levels of phosphatidylinositol-4,5-biphosphate, shows an increase in abundance after 24 hr of TTX treatment. We also observed corresponding elevated levels of phosphatidyl-4- and -5-phosphate-kinase subunits (Pip4k2b and Pip5k1c) and the related phosphatase (Inpp4a), as well as the AP2 related kinase 1 (Aak1) which phosphorylates AP-2 to trigger clathrin assembly. Some of the patterns of regulation complicate simple linear interpretations of signaling pathways: some regulated proteins that were downstream targets of a regulated kinase/phosphatase were often potential interactors with more than one regulated kinase/phosphatase. For example, the regulated protein Pdpk1, itself a kinase, interacts with three other regulated kinases (Lyn, Src and Fyn) as well as a regulated phosphatase (Ppp2r5e), many of which also interact with one another (*Figure 7A*). In addition, we observed that the abundance of a group of mitochondrial ribosomal protein subunits increased after both 2 hr of Bic and TTX treatment (Mrpl9, 19, 21 and 42; Mrps26) but then displayed different trends after 24 hr (*Figure 7F*).

Lastly, we investigated the regulated proteins that are known to possess the highest number of interaction partners. In this String analysis we acknowledge that we can only capture those interactions that are documented (e.g. for well-studied proteins) and thus could miss highly interactive proteins that have not been studied in detail. With this caveat in mind, we used the String database and identified the top ~40 regulated proteins that have the highest number of protein-protein interactions (*Figure 7G*). This group of proteins exhibits on average, 38.3 interactions with other proteins whereas a random sampling of 40 proteins exhibits on average 5.6 interactions. This difference is statistically significant (p<0.001). Not surprisingly, several kinases emerge (e.g. Adrbk1, Prkaca, Prkacb, Lyn, Kit) as well as phosphatases (Ptpn1, Ptpn11). We also observed several proteins involved in synaptic vesicle cycling (the syntaxins, Stx1a, Stx5, Stx6) and interestingly, proteins associate with the nuclear pore complex (Nup98 and Ranbp2). The regulation of proteins that have many interaction partners obviously can exert a multiplicative effect on downstream signaling during synaptic scaling.

## Discussion

Here we examined the proteomic response of neuronal networks to a brief but global manipulation (2 hr) of neuronal activity- either enhanced or reduced- by the addition of TTX or bicuculline. Using

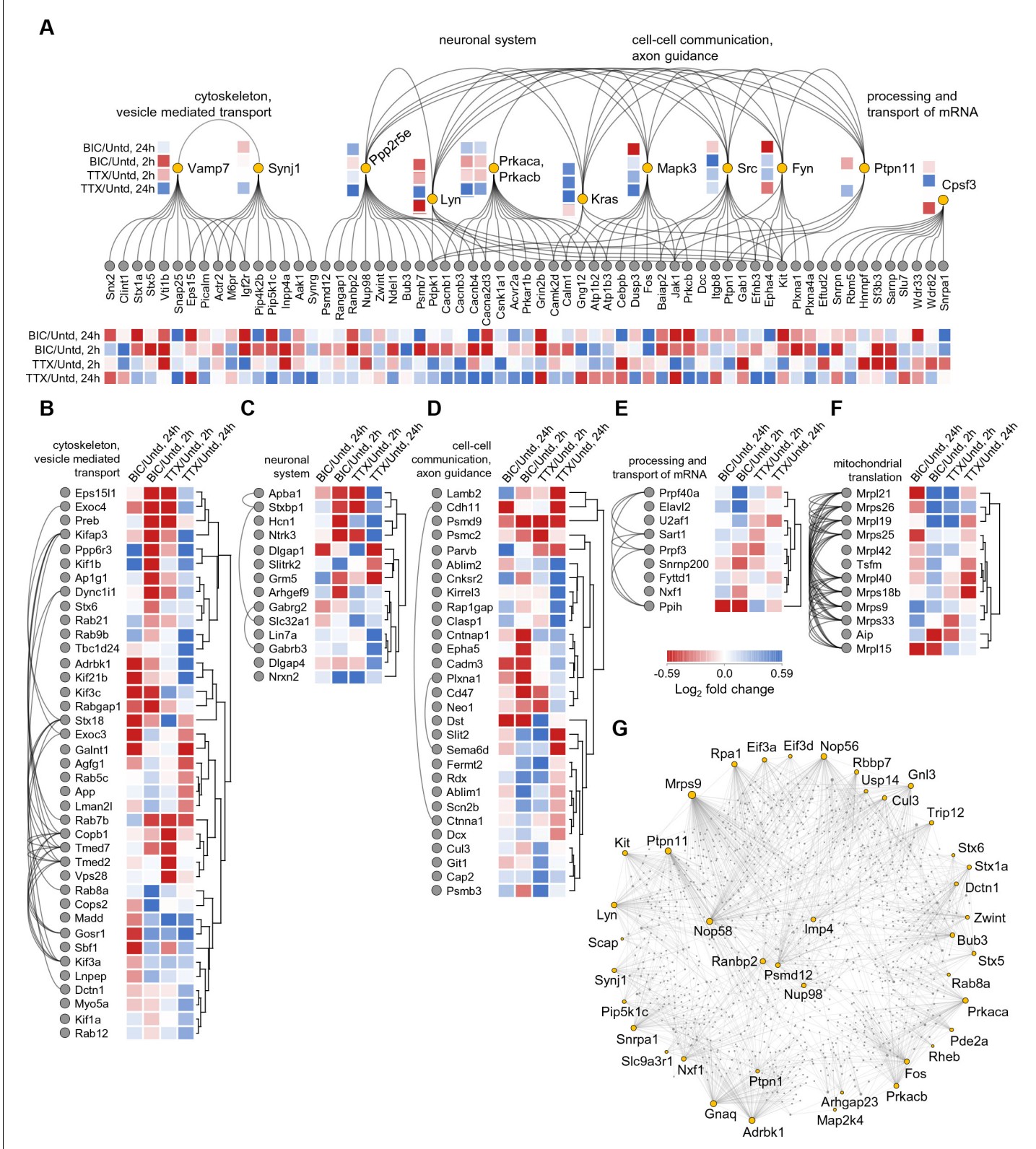

**Figure 7.** Network analysis of time- and polarity-dependent proteome regulation. (A–G) Using the String database, we analyzed the patterns of regulation and interaction between regulated proteins, using all 711 regulated proteins. (A) Here we indicate by functional groups, the most interactive proteins and protein groups including (B) cytoskeleton and vesicle-mediated transport, (C) neuronal system, (D) cell-cell communication and axon

*Figure 7 continued on next page*

*Figure 7 continued*

guidance, (**E**) processing and transport of RNA, and the highly reciprocally interactive group of regulated proteins (**F**) associated with mitochondrial translation. (**G**) The regulated proteins that function as highly interactive hubs in the proteome are depicted.

DOI: https://doi.org/10.7554/eLife.33322.012

BONCAT to metabolically label and identify the nascent proteome, we discovered 168 proteins whose expression levels were significantly regulated by treatments that lead to homeostatic up- or down-scaling. The proteins that we identified are largely associated with neuronal function, including ion channels, adhesion and guidance molecules, synaptic receptors, and synaptic vesicle-associated proteins. We also found that a large number of differentially regulated proteins are involved in post-translational modifications. Adjusting proteome composition and activity by modifying existing protein pools is an energy-efficient and rapid way to alter activity states. Following a 2 hr manipulation of activity, we observed elevated synthesis of several kinases such as Lyn, Aak1 and Src, and phosphatases such as protein tyrosine phosphatases (Ptprg, see *Figure 2* and *Figure 3*). We also observed the regulation of proteins involved with protein synthesis and degradation including Cullin3, an essential component of the E3 ubiquitin-protein ligase complexes essential for poly-ubiquitination and protein degradation. The regulation of degradation-related proteins supports observations that protein degradation plays an important role to adjust the neuronal proteome to the desired composition and thus functionality (e.g. *Bingol and Schuman, 2006*; *Tai and Schuman, 2008*). Lastly, while we identified several proteins (e.g. Rheb, Rgs2, Ppp1r7, and Cacnb3) that have been implicated in previous studies of homeostatic scaling, we did not observe significant regulation of AMPA receptors, which are known to be end-point effectors for both up- and down-scaling elicited after short-term manipulations of activity (*Ibata et al., 2008*). It is possible that our measurements were not sensitive enough to detect changes in AMPAR after a short (2 hr) metabolic labeling and activity manipulation. We did, however, observe polarized regulation of some AMPAR after 24 hr of labeling and manipulation (*Schanzenbächer et al., 2016*). This indicates that synaptic scaling following short-term activity manipulations may be due to post-translational modifications rather than regulation of new AMPAR synthesis.

We compared the regulated nascent proteome after 2 hr of global activity manipulation to a previous data set (*Schanzenbächer et al., 2016*) in which the same activity manipulations were applied for 24 hr. Surprisingly, there was very little overlap (~10 proteins) between the two time points. We noted two additional interesting features. First, a prominent feature of both datasets is the regulation of the same protein population by both up- and down-scaling manipulations. In some cases, the regulation of this common protein pool is polarized (e.g. upregulated in one condition and downregulated in another; *Figure 3*) reflecting the sign of the manipulation. In other cases, the polarity of the regulation is similar for both manipulations. Second, we analyzed if the fraction of overlapping regulated proteins was similar at 2 and 24 hr and found that this overlap was reduced from 52% at 2 hr to 34% at 24 hr. These results suggest that the longer the neuronal system is in a state of offset from a threshold, the more distinct the proteomic response becomes to reflect whether up- or down-scaling must be implemented.

From a network perspective, the regulated proteins identified here and in our previous study (*Schanzenbächer et al., 2016*) can function as different types of sensors and effectors during homeostatic scaling. The polarity (positive or negative) of the offset from an activity setpoint could be coded for by proteins uniquely synthesized in response to TTX or bicuculline treatment. In our experiments, the number of such unique proteins was low as was their expression level. Another means to represent the sign of the offset is the differential regulation (up- or down) of a single protein or multiple proteins. We detected several protein examples that fit this profile. Using self-organizing maps, we clustered the patterns of regulation observed over time and with each activity manipulation. Interestingly, we found that many patterns of regulation indicated the duration of the activity offset (2 vs 24 hr), but not the polarity (whether activity was enhanced or decreased). We also found several regulation patterns that indicate the polarity of the manipulation, showing changes in expression exclusively to one manipulation but not another. The proteins that do not show polarized regulation may represent general offset detectors whereas those that exhibit polarized responses may be effectors for the polarized homeostatic response. Taken together, these data

indicate that there are patterns of individual protein regulation that can convey information about either the duration of the activity offset (time) or the polarity (up-scaling or down-scaling) of the offset, or in some cases, both.

Capitalizing on the sensitivity of BONCAT, we were able to monitor the regulation of the neuronal proteome following a relatively brief global activity manipulation. In contrast to other recent publications (*Bowling et al., 2016*), we did not use SILAC-co-incorporation because the stringent biochemical purification method we have developed allows one to directly monitor newly-synthesized proteins. In addition, because of the high number of proteins identified, we did not need to make use of intensity-based thresholds for including proteins also detected at lower levels in control samples. We note, however, that AHA is incorporated into the proteomes of all cells present in the preparation, including glia cells, leading to a dilution of the measured neuronal proteome response analyzed in this study. We recently developed a technique that enables cell-type specific labeling and identification of nascent proteins (*Alvarez-Castelao et al., 2017*). Future studies can make use of this platform to monitor selectively the cell-type specific proteomes both in vitro and in vivo.

## Materials and methods

### Preparation of dissociated neuron cultures

Dissociated hippocampal neurons from postnatal day 0–2 rat pups (strain Sprague-Dawley) were prepared and maintained as previously described (*Aakalu et al., 2001*). Briefly, hippocampi were dissected and triturated after incubating in L-cystein-papain solution at 37°C for 15 min. Dissociated neurons were plated onto poly-D-Lysine-coated Petri dishes (MatTek) and cultured in Neurobasal A medium supplemented with B-27 and Glutamax (Invitrogen) at 37°C for 21 days. All experiments were carried out in accordance with the German Animal Welfare Act and supervised by the local government authorities and the Max Planck Society.

### Sample preparation

After a brief (30 min) methionine deprivation, two dishes of cultured neurons (~800 k cells in total) were incubated in each condition with 4 mM AHA (or methionine as a control) and treated with 20 µM bicuculline or 1 µM Tetrodotoxin (Tocris Bioscience) for 2 hrs. After incubation, the neurons were washed with cold DPBS completed with protease inhibitor cocktail (cOmplete EDTA-free, Roche), harvested, and pelleted at 2000x g for 5 min. The cell pellets were snap-frozen in liquid nitrogen, and stored at −80°C until further use.

The pelleted neurons were resuspended in lysis buffer (100 µL, 8 M urea, 200 mM Tris [pH 8.4], 4% CHAPS, 1 M NaCl, cOmplete EDTA-free protease inhibitor) and homogenized using a pestle. The lysates were sonicated with short bursts (4 × 30 s) in a cooled ultrasonic bath, followed by 5 min of benzonase digestion (1 µL of a $\geq$ 250 units/µL stock solution), and centrifugation for 5 min at 10000x g. The supernatants (~200 µL) were incubated with freshly prepared catalyst solution (250 µL) and Alkyne-Sepharose slurry (50 µL) according to the manufacturer's protocol (#C10416, Thermo Fisher Scientific). The reaction mixture was gently agitated for 19 hr in the dark. After centrifugation (2 min, 1000x g), the beads were rinsed twice in $H_2O$ (900 µL each) and incubated in 250 µL SDS wash buffer (100 mM Tris [pH 8], 1% SDS, 250 mM NaCl, 5 mM EDTA) containing 10 mM TCEP for 45 min at 55°C under gentle agitation. The mixture was centrifuged for 5 min at 1000x g and the supernatants were discarded.

Each sample was incubated with 250 µL SDS wash buffer containing 95 mM iodoacetamide for 30 min at room temperature in the dark under gentle agitation. The samples were transferred to pre-washed (400 µL $H_2O$, LC-MS/MS-ChromaSolv; 400 µL SDS wash buffer) Pierce Spin columns and then washed with 20 mL SDS wash buffer, 20 mL 8 M urea in 100 mM Tris [pH 8], and 20 mL 20% (v/v) acetonitrile/water. The resin was rinsed in digestion buffer (250 µL, 100 mM Tris, 2 mM $CaCl_2$, 10% acetonitrile) and transferred to an Eppendorf tube. After centrifugation (5 min, 1000x g), the supernatant was discarded leaving a volume of ~50 µL slurry. EndoLysC (0.65 µg) was added to each tube and incubated overnight (~22 hr) at 37°C with constant agitation. For subsequent tryptic digestion, each sample was incubated with 0.65 µg trypsin at 37°C overnight (~22 hr) with constant agitation. The samples were rinsed twice with 500 µL $H_2O$ (0.1% TFA) and centrifuged at 1000x g for 5 min. The collected supernatants were loaded onto $C_{18}$-SepPak columns (50 mg sorbent, Waters

Corp.) conditioned with 2 mL acetonitrile, 1 mL of 50% acetonitrile/water (0.5% acetic acid) in 0.5% acetic acid, and 2 mL water (0.1% TFA). The samples (~1 mL) were washed with 2 mL of water (0.1% TFA) and 200 µL of water (0.5% acetic acid). Desalted peptides were eluted with 0.5 mL of 50% acetonitrile/water (0.5% acetic acid), dried using a Speed-Vac (Eppendorf), and stored at −80°C until LC-MS/MS analysis.

## LC-MS/MS analysis

The dried peptide fractions were dissolved in 5% acetonitrile with 0.1% formic acid, and subsequently loaded using a nano-HPLC (Dionex U3000 RSLCnano) onto a PepMap100 trapping column ($C_{18}$, particle size 3 µm, L = 20 mm). Peptides were separated on a PepMap RSLC analytical column ($C_{18}$, particle size <2 µM, L = 50 cm, Dionex/Thermo Fisher Scientific) by a gradient of water (buffer A: water with 5% v/v dimethylsulfoxide and 0.1% formic acid) and acetonitrile (buffer B: 5% dimethylsulfoxide, 15% water and 80% acetonitrile (v/v/v), and 0.08% formic acid), running from 4% to 48% B in 178 min at a flowrate of 300 nL/min. All LC-MS-grade solvents were purchased from Fluka.

Peptides eluting from the column were ionized online using a Thermo nanoFlex ESI source and analyzed in a 'Q Exactive Plus' mass spectrometer (Thermo Fisher Scientific). Mass spectra were acquired over the mass range 350–1,400 m/z, and sequence information were acquired by data-dependent automated switching to MS/MS mode using collision energies based on mass and charge state of the candidate ions (TOP12, MS resolution 70 k, MS/MS resolution 35 k, injection time: 120 ms, full parameters in *Supplementary file 8*). All samples were measured in quadruplicate LC-MS/MS runs.

## Data analysis

MS data were analyzed in MaxQuant (ver. 1.5.5.1; *Cox and Mann, 2008*) using a customized Andromeda LFQ parameter set (see *Supplementary file 8*). In brief, spectra were matched to a *Rattus norvegicus* database downloaded from uniprot.org (35,953 entries, reviewed and non-reviewed) and a contaminant and decoy database. Tryptic peptides with ≥6 amino acids and ≤2 missed cleavages were included. Precursor mass tolerance was set to 4.5 ppm, fragment ion tolerance to 20 ppm, with a static modification of Cys residues (carboxyamidomethylation +57.021) and variable modifications of Met residues (oxidation +15.995), Lys residues (acetylation +42.011), Asn and Gln residues (deamidation +0.984), and N termini (carbamylation +43.006). Search results were filtered with an FDR of 0.01, and proteins identified by ≥1 unique peptide were included for subsequent analysis. Proteins were quantified in each condition by pair-wise ratio determination using ≥1 common peptide in ≥4 consecutive full scans per run for the label-free quantification (*Cox et al., 2014*).

## Bioinformatic processing

Five independent biological replicates measured in quadruplicates were processed in Perseus software package (ver. 1.5.5.3; *Tyanova et al., 2016*). After combining technical replicates, background proteins detected in vehicle control were subtracted from the other conditions (untreated, bicuculline-treated, TTX-treated neurons). Newly-synthesized proteins identified in each biological replicate were averaged within each condition (*Figure 1A*, *Supplementary file 1*), and counted according to their numbers of biological replicates (*Figure 1—figure supplement 1A*). Venn diagrams (e.g. showing the protein overlap in ≥1 biological replicate, *Figure 1B*) were generated using Venn Diagram Plotter (ver. 1.5.5228.29250; PNNL). Sub-cellular localization of proteins identified in ≥2 biological replicates (*Figure 1—figure supplement 1B*) was predicted using the LocTree3 database (ver. 23-08-2016; *Goldberg et al., 2014*).

Non-normalized protein intensities (non-LFQ) were $log_2$ transformed and averaged across replicates to assess differences in overall protein abundances (*Figure 1—figure supplement 1C*, *Supplementary file 1*). LFQ intensities were normalized over experimental conditions using the central tendency adjustment method: protein quantities were divided by a normalization factor representing the median intensity of all proteins measured in ≥2 biological replicates. The normalized $log_2$ intensities were averaged across ≥2 biological replicates and then used for calculating the fold change of treated neurons relative to control (*Figure 1C*, *Supplementary file 1*).

After filtering for the presence in ≥2 biological replicates (3196 proteins remaining) and for a coefficient of variation <1 (3086 proteins remaining), these proteins were annotated using the gene

ontology databases (GOBP, GOMF, GOCC, *Ashburner et al., 2000*), the Panther database (*Thomas et al., 2003*) and protein keywords retrieved from the UniprotKB database (*The UniProt Consortium, 2017*). Differentially regulated groups of annotated proteins were analyzed and extracted using Perseus (*Figure 3*, *Supplementary file 4*, 1D annotation enrichment, two-sided Wilcoxon-Mann-Whitney test, Benjamini-Hochberg FDR of 0.05), and compared to the protein fold changes after 24 hr (*Figures 4G*, 24 hr dataset from *Schanzenbächer et al., 2016*). Protein annotations of both time points were then combined and analyzed in a 2D annotation enrichment (*Cox and Mann, 2012*) using a p value < 0.01 (*Figure 4H*, *Supplementary file 4*). Duplicates and annotation groups with more than 100 protein members were discarded.

Statistical significance for protein regulation between bicuculline/untreated, TTX/untreated or bicuculline/TTX after 2 hr were measured using an ANOVA (permutation-based FDR of 0.05, $S_0 = 0.05$, 250 randomizations), followed by post-hoc Fisher LSD $p<0.05$ (in Origin 2015G Sr1, OriginLab), yielding 168 proteins (*Figure 1D*, *Figure 2C*, *Supplementary file 2*). Significantly regulated proteins were annotated (*Figure 2A*, *Supplementary file 3*) using the VarElect interpretation tool (*Stelzer et al., 2016*) by LifeMap's GeneCards suite (*Ben-Ari Fuchs et al., 2016*). Cellular component ontologies enriched in the regulated proteome compared to the 5135 newly synthesized proteins as the background were analyzed in FunRich (ver. 3; *Pathan et al., 2015*) using a Benjamini Hochberg corrected p value < 0.05 (*Figure 2B*, *Supplementary file 3*). Significantly regulated proteins in early (2 hr) and late response (24 hr, *Schanzenbächer et al., 2016*) were compared (*Figure 4B*, *Figure 4—figure supplement 1B*), and the protein overlap regulated in both treatments was calculated for each time point using Fisher LSD $p<0.05$ (*Figure 4C–D*, *Supplementary file 2* and *5*). The 'degree of regulation' was calculated for the 168 and 307 regulated proteins (2 hr and 24 hr, respectively) by measuring the distance of each protein to the zero point in *Figure 4E–F*.

Protein fold changes of treated neurons relative to the control were calculated for each biological replicate in both time points. Time- and polarity-dependent regulation was statistically validated by an ANOVA (permutation-based FDR of 0.05, $S_0 = 0.05$, 250 randomizations), yielding 711 differentially regulated proteins (*Supplementary file 6*).

Cell-type specific markers were extracted from *Sharma et al., 2015* by selecting 290 glial cell and 644 neuronal markers showing an at lest 2-fold higher abundance in astrocytes, oligodendrocytes and microglia cultures compared to neuronal cultures. Statistical significance of neuronal markers enriched in the differentially regulated proteomes were evaluated by comparing the number of regulated neuronal and glial markers. Protein clusters (*Figure 6*, *Figure 6—figure supplement 1B*, *Supplementary file 7*) were compiled in J-Express pro 2012 (*Dysvik and Jonassen, 2001*) using a Self-Organizing Map (SOM) algorithm (49 nodes, theta/momentum = 0.998, phi/momentum = 0.998, Euclidean distance, Gaussian neighbourhood function, ≤4000 iterations). SOM clusters were hierarchically clustered in a second layer and assembled in a heatmap (*Figure 5*, Uncentered Pearson Correlation, Weighted Average Linkage, WPGMA). Enrichment analysis of cellular component and molecular function ontologies (*Figure 6—figure supplement 1B*, *Supplementary file 7*) was conducted in FunRich (Benjamini Hochberg corrected p value < 0.05).

Protein networks (*Figure 7A–F*) were analyzed using the String interactome (v. 10.0, string-db.org, *Szklarczyk et al., 2015*) with databases only as interaction source and high confidence of interactions (score >0.700). Protein groups (*Figure 7F*) were preprocessed with k-means and hierarchically clustered in Perseus (Euclidean distance, complete linkage). Significantly regulated proteins that are annotated as 'high interaction hubs' (*Figure 7G*) were analyzed in NetworkAnalyst (networkanalyst.ca, *Xia et al., 2015*) using the String interactome (confidence score >0.900, experimental evidenced, first-order network). The largest network containing 82 regulated proteins, 1178 interaction partners and 1745 protein-protein interactions was plotted in *Figure 7G* using the force atlas layout algorithm.

The proteomics data associated with this manuscript have been deposited to the ProteomeXchange Consortium via the PRIDE partner repository (*Vizcaíno et al., 2016*) with the dataset identifier PDX008271.

## Acknowledgements

We thank I Bartnik, N Fuerst, A Staab and C Thum for the preparation of cultured hippocampal cells. EMS is funded by the Max Planck Society, an Advanced Investigator award from the European

Research Council, and DFG CRC 1080: Molecular and Cellular Mechanisms of Neural Homeostasis, DFG CRC 902: Molecular Principles of RNA-based Regulation and the DFG Cluster of Excellence for Macromolecular Complexes, Goethe University.

## Additional information

### Funding

| Funder | Grant reference number | Author |
|---|---|---|
| Max-Planck-Gesellschaft | Open-access funding | Erin M Schuman |

The funders had no role in study design, data collection and interpretation, or the decision to submit the work for publication.

### Author contributions

Christoph T Schanzenbächer, Conceptualization, Supervision, Funding acquisition, Visualization, Methodology, Writing—original draft, Project administration; Julian D Langer, Conceptualization, Data curation, Formal analysis, Validation, Investigation, Visualization, Methodology, Writing—review and editing; Erin M Schuman, Conceptualization, Data curation, Formal analysis, Supervision, Investigation, Project administration, Writing—review and editing

### Author ORCIDs

Julian D Langer (iD) https://orcid.org/0000-0002-5190-577X
Erin M Schuman (iD) http://orcid.org/0000-0002-7053-1005

### Ethics

Animal experimentation: We hereby confirm that all the experiments involving animals (i.e. postmortem tissue removal as defined in the § 4(3) of German animal welfare act) that were done in relation to our manuscript entitled "Time- and Polarity-dependent Proteomic Changes associated with Homeostatic Scaling at Central Synapses" were carried out in accordance with the European directive 2010/63/EU, the German animal welfare act, and the guidelines of the Federation of Laboratory Animal Science Associations (FELASA) and the Max Planck Society. All rats were maintained at the Max Planck Institute for Brain Research animal facility (Frankfurt, Germany). Maternal Sprague-Dawley rats (Crl:CD(SD), Charles River Laboratories, Sulzfeld, Germany) were received at day 18 after plug observation (timed matings at Charles River Laboratories) and were kept in the animal facility until litter delivery at day 23. The maternal rats were housed under specified-pathogen-free conditions in Tecniplast Doppeldecker cages at 20-24°C room temperature, with room lighting set to a 12:12 hour light-dark cycle. Rats received a commercial diet (ssniff Spezialdiäten, Soest, Germany) and water ad libitum. Cage bedding was a sterilized commercial softwood granulate (Lignocel BK 8-15, J. Rettenmaier & Söhne, Rosenberg, Germany). Dissociated rat hippocampal neurons were prepared from 1-day-old Sprague-Dawley rat pups that were sacrificed, immediately after removal from the mother, by decapitation with sharp scissors.

### Decision letter and Author response

Decision letter https://doi.org/10.7554/eLife.33322.025
Author response https://doi.org/10.7554/eLife.33322.026

## Additional files

### Supplementary files

• Supplementary file 1. Complete lists of newly-synthesized proteins at 2 h and 24 h.
DOI: https://doi.org/10.7554/eLife.33322.013

• Supplementary file 2. ANOVA of polarity-dependent protein regulation after 2 hrs and previously implicated proteins.

DOI: https://doi.org/10.7554/eLife.33322.014

• Supplementary file 3. Protein classification and GO enrichment analysis.
DOI: https://doi.org/10.7554/eLife.33322.015

• Supplementary file 4. Statistical analysis of differentially regulated functional groups.
DOI: https://doi.org/10.7554/eLife.33322.016

• Supplementary file 5. Significantly regulated proteins in late response (24 hrs) and protein overlap in both timepoints.
DOI: https://doi.org/10.7554/eLife.33322.017

• Supplementary file 6. ANOVA of time- and polarity-dependent protein regulation and marker enrichment.
DOI: https://doi.org/10.7554/eLife.33322.018

• Supplementary file 7. Protein clustering and GO enrichment analysis.
DOI: https://doi.org/10.7554/eLife.33322.019

• Supplementary file 8. Full LC-MS and data analysis parameters.
DOI: https://doi.org/10.7554/eLife.33322.020

• Transparent reporting form
DOI: https://doi.org/10.7554/eLife.33322.021

### Major datasets

The following dataset was generated:

| Author(s) | Year | Dataset title | Dataset URL | Database, license, and accessibility information |
|---|---|---|---|---|
| Schanzenbächer C, Langer J, Schuman E | 2018 | Time- and Polarity-dependent Proteomic Changes associated with Homeostatic Scaling at Central Synapses. | https://www.ebi.ac.uk/pride/archive/projects/PXD008271 | Publicly available at EBI PRIDE (accession no. PXD008271) |

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
