## [Decision Letter]

Thank you for submitting your article "Time- and Polarity-dependent Proteomic Changes associated with Homeostatic Scaling at Central Synapses" for consideration by *eLife*. Your article has been favorably evaluated by a Senior Editor and three reviewers, one of whom, Yukiko Goda (Reviewer #1)., is a member of our Board of Reviewing Editors.

The reviewers have discussed the reviews with one another and the Reviewing Editor has drafted this decision to help you prepare a revised submission

Schanzenbacher et al. have examined the proteomic changes associated with treatments that elicit homeostatic synaptic scaling in cultured hippocampal neurons, using BONCAT nascent peptide-labeling combined with MS and MaxQuant data analysis methods. Following their previous work describing differentially expressed proteins after 24 h of activity manipulation (Neuron 2016), the authors now add a 2 h time point and compare the dynamics of the proteome across time. They find that many of the regulated proteins are associated with neuronal and/or synaptic functions and show distinct behaviors with respect to time and polarity of activity manipulation. Curiously, there is very little overlap of the regulated expression of individual proteins between 2 h and 24 h of activity manipulation despite the involvement of common protein function categories across both time periods. The findings suggest a key role for these functional categories in homeostatic scaling and, potentially, a time-dependent specialization of the regulation of specific proteins belonging to the same categories. Altogether, the results appear robust and should of broad interest to the community while providing a valuable resource. However, some revisions not involving additional experiments are requested as appended below.

Essential revisions:

1) There is very low correlation between the untreated 2hrs data and the untreated 24hrs data. Is this due to the difference in the duration of the BONCAT labeling? The authors state "In the absence of activity manipulation, the 170 newly synthesized proteomes identified at these two time points were largely (~94%) 171 overlapping (Figure 4 and Figure 4—figure supplement 1)". But in Figure 1—figure supplement 1, the correlation coefficient between 2h untreated and 24h untreated is 0.69, that is much lower than that of treated vs. untreated (>0.95). This means the difference between 2h untreated and 24h untreated is much bigger than the difference between treated and untreated of same duration. Why are the proteins synthesized in neurons within 2h so different from those within 24h? This difference may severely bias their comparison of the early- and late-response to an activity manipulation and requires further explanation.

2) The authors use the term 'hippocampal neurons' throughout the manuscript (e.g. the Abstract), including the Materials and methods, without qualifying that the cultures are a mix of neurons and glia. It therefore comes as a surprise to learn at the end of the Discussion "that AHA is incorporated into the proteomes of all cells present in the preparation, including glia cells, leading to a dilution of the measured neuronal proteome response analyzed in this study." The authors should discuss the implications of the presence of glia in their samples up-front and in more detail. What is the ratio of neurons to glia in the cultures? What different types of hippocampal neurons are present? Since it is not accurate to use the term 'hippocampal neurons' to describe the biological preparation, the authors should either change it to 'hippocampal cells' or qualify the use of the term early in the manuscript.

3) The manuscript would benefit from a more in-depth Discussion to expand the scope of the present findings with respect to current understanding of the molecular mechanisms of homeostatic scaling. In particular, the following points need clarifications.

a) Why the majority of proteomic changes with greater magnitudes show little polarity despite the fact that some do show opposing polarity with directional changes in network activity?

- Data in Figure 4 is rather informative. At both 2h and 24h time points, more data points fall into the upper right or lower left quadrants, and that these data points have overall a bigger deviation from the origin, indicating greater changes. From this, one would conclude that most proteomic changes that occur doing scaling with greater magnitudes have no polarity, and reflect some basic cellular processes that are engaged when prolonged shift in neural network/synaptic activity occur. What is really surprising is that some of these protein groups (for example, dendritic spine, synapse, exocytosis etc., Figure 4) have a clear polarized response in scaling; they change in opposite directions during up and down scaling. What might be the significance for such behaviors?

b) Why AMPARs show no changes at 2h of activity manipulation while at 24 h, they show changes in the same direction irrespective of the direction of activity manipulation?

- Surprisingly AMPAR complex has no significant proteomic response at 2h (Figure 6—figure supplement 1), while a previous report (Ibata et al., 2008) showed that TTX can rapidly induce scaling within 2h. Additionally, at 24h, AMPAR complex changes to the same direction with both up and down scaling treatments (Figure 6—figure supplement 1), while the synaptic changes mediated by AMPAR during up and down scaling are clearly opposite. How do the authors explain such discrepancy?

---

## [Author Response]

Essential revisions:1) There is very low correlation between the untreated 2hrs data and the untreated 24hrs data. Is this due to the difference in the duration of the BONCAT labeling? The authors state "In the absence of activity manipulation, the 170 newly synthesized proteomes identified at these two time points were largely (~94%) 171 overlapping (Figure 4 and Figure 4—figure supplement 1)". But in Figure 1—figure supplement 1, the correlation coefficient between 2h untreated and 24h untreated is 0.69, that is much lower than that of treated vs. untreated (>0.95). This means the difference between 2h untreated and 24h untreated is much bigger than the difference between treated and untreated of same duration. Why are the proteins synthesized in neurons within 2h so different from those within 24h? This difference may severely bias their comparison of the early- and late-response to an activity manipulation and requires further explanation.

We thank the reviewer(s) for bringing up this point since it is important to understand. The differences in the “control” (unstimulated) proteomes at 2 hr and 24 hr represent differences in metabolic labeling efficiency. Simply put, the presence of AHA for 24 hrs gives us a much greater opportunity to capture the synthesis of a given protein. The overlap is protein identity is indeed high (94%) but the correlation in Figure 1—figure supplement 1 reflects the protein amount, not the identity, only for those proteins that are detected at both time points (e.g. the 171 overlapping proteins). It is important to note that, even though overall intensities went up in 24h, we can still track and quantify the same proteins and compare the plasticity effects at 24 h and 2h- this is because each time point is normalized to its own (same) time point control. We have expanded on our explanation of the Figure S1D legend in order to make this point more clear.

2) The authors use the term 'hippocampal neurons' throughout the manuscript (e.g. the Abstract), including the Materials and methods, without qualifying that the cultures are a mix of neurons and glia. It therefore comes as a surprise to learn at the end of the Discussion "that AHA is incorporated into the proteomes of all cells present in the preparation, including glia cells, leading to a dilution of the measured neuronal proteome response analyzed in this study." The authors should discuss the implications of the presence of glia in their samples up-front and in more detail. What is the ratio of neurons to glia in the cultures? What different types of hippocampal neurons are present? Since it is not accurate to use the term 'hippocampal neurons' to describe the biological preparation, the authors should either change it to 'hippocampal cells' or qualify the use of the term early in the manuscript.

We have changed the wording to indicate hippocampal cells instead of hippocampal neurons. We estimate that in our mixed hippocampal cell cultures we have about 75% neurons and 25% glia. In addition, we have now examined a set of glial and neuronal cell marker proteins derived from a study from Matthias Mann’s lab (Sharma et al) and identified 644 and 290 neuronal and glial marker proteins, respectively. When we analyzed how many of the 711 regulated proteins we identified were neuronal or glial markers, we found 88 and 7, respectively. There is a clear statistically significant enrichment for neuronal markers. These new analyses are shown in new Supplementary file 6.

3) The manuscript would benefit from a more in-depth Discussion to expand the scope of the present findings with respect to current understanding of the molecular mechanisms of homeostatic scaling. In particular, the following points need clarifications.a) Why the majority of proteomic changes with greater magnitudes show little polarity despite the fact that some do show opposing polarity with directional changes in network activity?- Data in Figure 4 is rather informative. At both 2h and 24h time points, more data points fall into the upper right or lower left quadrants, and that these data points have overall a bigger deviation from the origin, indicating greater changes. From this, one would conclude that most proteomic changes that occur doing scaling with greater magnitudes have no polarity, and reflect some basic cellular processes that are engaged when prolonged shift in neural network/synaptic activity occur. What is really surprising is that some of these protein groups (for example, dendritic spine, synapse, exocytosis etc., Figure 4) have a clear polarized response in scaling; they change in opposite directions during up and down scaling. What might be the significance for such behaviors?

We agree with the reviewer that this is a very interesting issue – the observation that more regulated proteins show expression changes in the same direction for both up- and down-scaling and that some proteins (which one might speculate are proximal to the expression of synaptic scaling, for example) do exhibit polarized responses as one would expect if they are effectors. We have expanded our discussion of this in the revised manuscript (Discussion section).

b) Why AMPARs show no changes at 2h of activity manipulation while at 24 h, they show changes in the same direction irrespective of the direction of activity manipulation?- Surprisingly AMPAR complex has no significant proteomic response at 2h (Figure 6—figure supplement 1), while a previous report (Ibata et al., 2008) showed that TTX can rapidly induce scaling within 2h. Additionally, at 24h, AMPAR complex changes to the same direction with both up and down scaling treatments (Figure 6—figure supplement 1), while the synaptic changes mediated by AMPAR during up and down scaling are clearly opposite. How do the authors explain such discrepancy?

We thank the reviewer(s) for bringing up this point so that we can clarify any uncertainty or misconceptions. The Ibata report of synaptic scaling does not require an increase in the basal levels of AMPAR synthesis. Changes in synaptic transmission that involve or require AMPA receptors can be due to differences in post-translation modifications that change the number, position, dwell-time at synapses, current, etc. Our detection of changes in AMPAR at 24 hrs could also be upstream of the posttranslational modifications that give polarity to the scaling. We have expanded our discussion of this topic in the revised manuscript.